# HIF1α is required for NK cell metabolic adaptation during virus infection

Francisco Victorino[1]*, Tarin M Bigley[1], Eugene Park[1], Cong-Hui Yao[2], Jeanne Benoit[3], Li-Ping Yang[1], Sytse J Piersma[1], Elvin J Lauron[1], Rebecca M Davidson[3], Gary J Patti[2], Wayne M Yokoyama[1]*

[1]Rheumatology Division, Washington University School of Medicine, St. Louis, United States; [2]Department of Chemistry, Department of Medicine, Washington University, St. Louis, United States; [3]Department of Biomedical Research, Center for Genes, Environment and Health, National Jewish Health, Denver, United States

**Abstract** Natural killer (NK) cells are essential for early protection against virus infection and must metabolically adapt to the energy demands of activation. Here, we found upregulation of the metabolic adaptor hypoxia-inducible factor-1α (HIF1α) is a feature of mouse NK cells during murine cytomegalovirus (MCMV) infection in vivo. HIF1α-deficient NK cells failed to control viral load, causing increased morbidity. No defects were found in effector functions of HIF1αKO NK cells; however, their numbers were significantly reduced. Loss of HIF1α did not affect NK cell proliferation during in vivo infection and in vitro cytokine stimulation. Instead, we found that HIF1α-deficient NK cells showed increased expression of the pro-apoptotic protein Bim and glucose metabolism was impaired during cytokine stimulation in vitro. Similarly, during MCMV infection HIF1α-deficient NK cells upregulated Bim and had increased caspase activity. Thus, NK cells require HIF1α-dependent metabolic functions to repress Bim expression and sustain cell numbers for an optimal virus response.

**\*For correspondence:** ramirezvictorino@wustl.edu (FV); yokoyama@wustl.edu (WMY)

**Competing interest:** The authors declare that no competing interests exist.

## Introduction

During infection with a cytolytic virus, such as murine cytomegalovirus (MCMV), infected cells become sources of viral replication and succumb to viral cytopathology. Natural killer (NK) cells play an integral role in initiating and fine-tuning immune responses to provide protection against MCMV. NK cells carry out this role through two overlapping functional properties: (1) secretion of pro-inflammatory cytokines and (2) direct killing of target cells. MCMV elicits non-specific NK cell activation through a strong host cytokine response but can also trigger specific NK cell activation in C57BL/6 mice. In particular, the Ly49H activation receptor recognizes the viral encoded protein m157 on infected cells, triggering cytotoxicity and cytokine production that can modulate subsequent immune responses (*Arase et al., 2002*; *Biron and Tarrio, 2015*; *Mitrović et al., 2012*; *Pyzik and Vidal, 2009*; *Smith et al., 2002*). These non-specific and specific NK cell responses are integrated to provide optimal NK cell responses and protection (*Dokun et al., 2001*). Ly49H+ NK cells subsequently acquire properties characteristic of adaptive immunity of expansion, contraction, and generation of memory-like responses (*Sun et al., 2009*). Thus, NK cells function as sentries during MCMV infection, controlling pathogen propagation and inflammation through their effector functions.

During inflammation, rapid energy consumption by cells for activation, biosynthesis, and the generation of reactive oxygen species results in hypoxia, a reduction in oxygen levels, even in the presence of environmental normoxia. Hypoxia-inducible factor 1-α (HIF1α) is an evolutionarily conserved transcription factor whose protein expression is determined in an oxygen-dependent manner (*Graham and Presnell, 2017*; *Kaelin, 2018*; *Schofield and Ratcliffe, 2004*; *Semenza, 2014*). Briefly, when

oxygen is readily available, hydroxylation of target residues on HIF1α results in ubiquitination and proteasome degradation while, during hypoxia this modification does not occur, allowing for protein accumulation and translocation into the nucleus. Under hypoxic conditions, HIF1α targets genes that contain hypoxia-response elements (HREs) to initiate transcriptional changes for adaption, supporting multiple cellular processes such as cell proliferation, survival, and glucose metabolism (*Dengler et al., 2014*; *Schödel et al., 2011*). Thus, HIF1α expression is directly associated with degree of oxygen levels and is necessary for adaption to hypoxia during inflammation.

Hypoxia is a fundamental characteristic of tumor environments and drives HIF1α protein expression in situ. HIF1α has been considered a potential mediator of suppressive effects in NK cells, which demonstrate impaired effector functions in hypoxia in vitro that mirror tumor-infiltrating NK cells (*Baginska et al., 2013*; *Balsamo et al., 2013*; *Chambers and Matosevic, 2019*). Indeed, deletion of HIF1α in NK cells resulted in an improved ability to reduce tumor burden in two independent studies, which was attributed to a dysfunctional VEGF axis or increased effector functions through IL-18 signaling (*Krzywinska et al., 2017*; *Ni et al., 2020*). HIF1α also supports metabolic adaptation to the tumor environment by upregulating key glycolysis genes to support cell proliferation and survival. Interestingly, Ni et al. showed that HIF1αKO NK cells had higher oxidative phosphorylation (OXPHOS) after cytokine activation, presumably to compensate for poor glucose metabolism in the absence of HIF1α upregulated genes. However, measurements of glucose metabolism showed no differences compared to control mice. Using similar in vitro cytokine stimulation experiments, another study showed that glucose metabolism in NK cells was regulated by cMyc, and HIF1α was found to be dispensable and the absence of HIF1α had no effect on OXPHOS metabolism (*Loftus et al., 2018*). Regardless, as a caveat, the pro-inflammatory cytokines used are more associated with pathogen infection rather than a tumor environment. Therefore, the contribution of HIF1α to NK cell metabolism is incompletely understood, particularly during the pathogen infection.

In this paper, we utilized in vitro assays and MCMV infections to study the role of HIF1α in NK cells. Compared to control mice, the absence of HIF1α in NK cells resulted in susceptibility to infection, but this was not due to an impaired effector response. Instead, we show a previously unknown role for HIF1α in NK cells during MCMV infection, supporting survival rather than proliferation through efficient glucose metabolism that is required for an optimal host response to virus infection.

## Results

### HIF1α is basally expressed in NK cells and is upregulated by MCMV infection

We found that HIF1α protein was readily detectable in splenic NK cells from naïve mice, immediately ex vivo using anti-HIF1α (*Figure 1A*). By contrast, there was essentially no HIF1α expression in other naïve lymphocytes directly ex vivo (*Figure 1B*). In addition, splenic and liver NK cells had similar HIF1α expression (*Figure 1—figure supplement 1A*). Under hypoxic conditions as compared to normoxia, HIF1α protein expression increased in NK cells cultured in vitro (*Figure 1C*). Interestingly, under normoxic conditions, IFNβ+ IL18 upregulated HIF1α expression to levels comparable to hypoxic conditions, though these cytokines did not further increase HIF1α expression under hypoxic conditions. Moreover, when we cultured splenocytes in IL-15 under normoxic conditions for 16, 24, and 48 hr, we found highest expression at 48 hr (*Figure 1D*). In contrast, HIF1α expression was not significantly upregulated after activation receptor stimulation (*Figure 1—figure supplement 1B*). Thus, NK cells constitutively express HIF1α immediately ex vivo that can be increased by hypoxia alone or cytokine stimulation in an oxygen-rich environment.

To determine HIF1α expression in NK cells during an immune response in vivo, we studied C57BL/6 mice infected with MCMV. At all doses of MCMV, splenic NK cell activation was observed on day 1.5 post-infection (pi) as indicated by CD69 expression, an activation marker not expressed on naïve NK cells (*Figure 1E*), and concomitant with increased HIF1α expression. HIF1α expression was highest at day 3 pi regardless of expression of the Ly49H receptor that activates and induces NK cell proliferation and memory formation (*Figure 1F*). CD69 expression was not associated with differences in HIF1α expression in NK cells day 3 post MCMV infection (*Figure 1—figure supplement 1C*). Thus, MCMV infection drives upregulation of HIF1α in NK cells, independent of activation receptor stimulation.

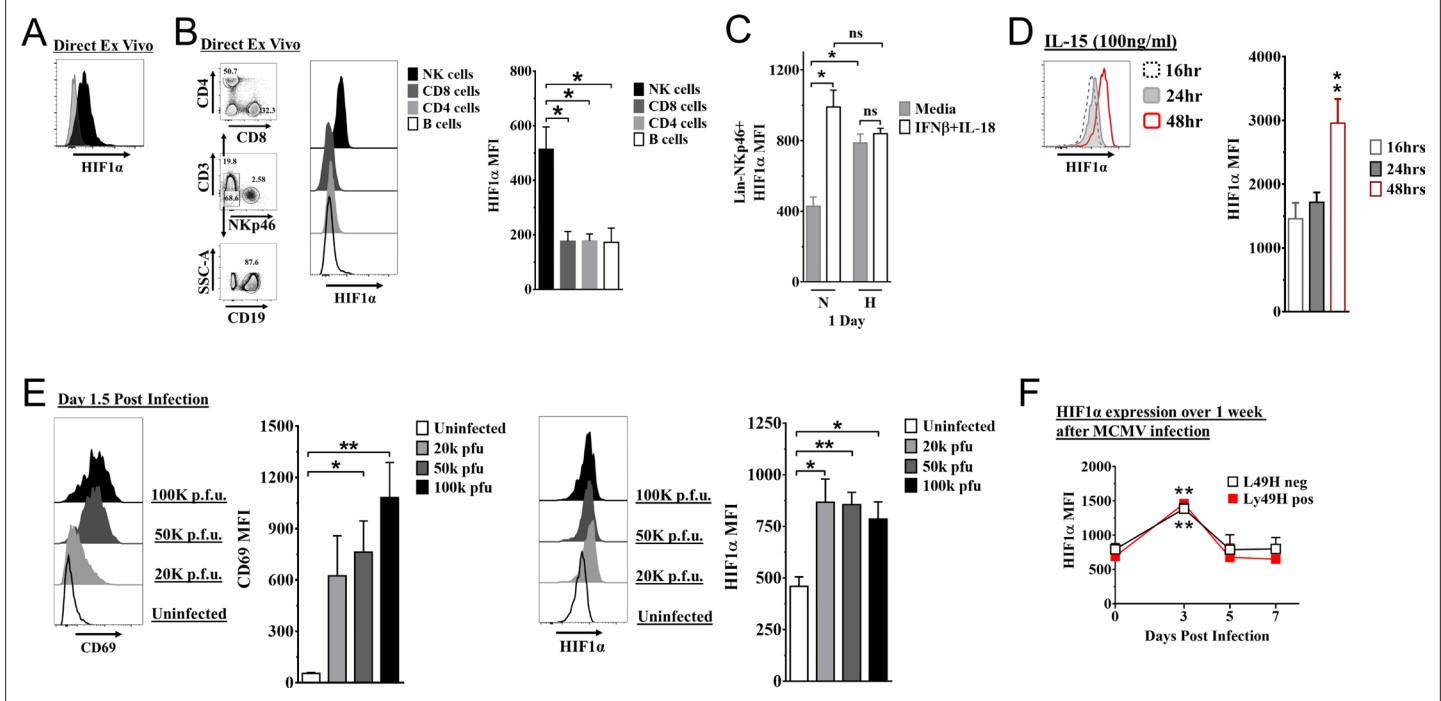

**Figure 1.** Hypoxia-inducible factor-1α (HIF1α) expression in natural killer (NK) cells. (**A**) HIF1α protein expression in splenic NK cells (black) ex vivo compared to FMO control (gray) by flow cytometry. Representative experiment of two independent experiments. (**B**) Comparison of lymphocytes HIF1α protein expression. Bar graphs show data from two independent experiments with four mice per group. (**C**) Spleens from C57BL/6 mice were prepared as in Materials and methods and were stimulated with IFNβ (200 units) + IL-18 (50 $\eta$ g/ml) for 24 hr in normoxia (N, 20%) or hypoxia (H, 1%). HIF1α expression in NK cells was determined. Data are from two independent experiments with three mice per group. (**D**) Splenic NK cells were cultured in 25 $\eta$ g/ml IL-15 for 48 hr and HIF1α expression was measured at 16, 24, and 48 hr. Data are from two independent experiments with four mice per group. (**E**) CD69 and HIF1α was determined at day 1.5 post-infection (pi). Data are from 2–3 independent experiments with three mice per group. Statistical significance for uninfected versus indicated dose. (**F**) HIF1α expression in Ly49H+ or Ly49H- NK cells over 7 days from C57BL/6 mice that were infected with 50 K plaque forming unit (pfu) murine cytomegalovirus (MCMV). Statistical significance for d0 versus indicated day. Data are from two independent experiments with 3–7 mice per group. All data depict mean ± SEM, with each data set containing data indicated number of mice per group from independent experiments. Unpaired t-test was performed (**A, B, D–F**) or one-way ANOVA (**C**). Statistical significance indicated by n.s., no significant difference; *p<0.05; **p<0.01; ***p<0.001; ****p<0.0001.

The online version of this article includes the following figure supplement(s) for figure 1:

**Figure supplement 1.** Expression of hypoxia-inducible factor-1α (HIF1α) in natural killer (NK) cells.

## NK cells require HIF1α for an optimal response to virus infection

To determine if HIF1α may contribute to NK cell function in vivo, we crossed *Ncr1*[iCre] mice with B6.129-*Hif1a<tm3Rsjo>*/J mice for NK cell-specific deletion of HIF1α (1αKO) for comparison to B6.129-*Hif1a<tm3Rsjo>*/J (FL) control mice that do not express Cre. The frequencies and numbers of NK cells in 1αKO mice were comparable to FL controls (*Figure 2—figure supplement 1A*) and there was normal NK cell development, as indicated by CD27 and CD11b expression profiles (*Figure 2— figure supplement 1B*). Expression of other functional markers and receptors, such as DNAM-1, KLRG1, NKG2A (*Figure 2—figure supplement 1C*), and the Ly49 repertoire, was also normal (*Figure 2—figure supplement 1D*). These data support previous findings that HIF1α is dispensable for normal NK cell development (*Krzywinska et al., 2017*).

During MCMV infection, however, we found that 1αKO mice showed increased weight loss as compared to control FL mice (*Figure 2A*). This increased morbidity was coupled with an impaired ability to control splenic viral load at day 5 pi (*Figure 2B*). The number of NK cells in 1αKO mice declined at day 1.5 pi and was significantly reduced at day 3 pi (*Figure 2C*), consistent with the kinetics of HIF1α expression in infected WT mice (*Figure 1E and F*). While FL control mice showed the characteristic expansion and contraction of Ly49H+ NK cells over 1 month of infection as previously reported (*Sun et al., 2009*), the 1αKO mice displayed reduced Ly49H+ NK cell expansion (*Figure 2D*). In addition, Ly49H- NK cells were also significantly impacted by the loss of HIF1α at day 3 and day

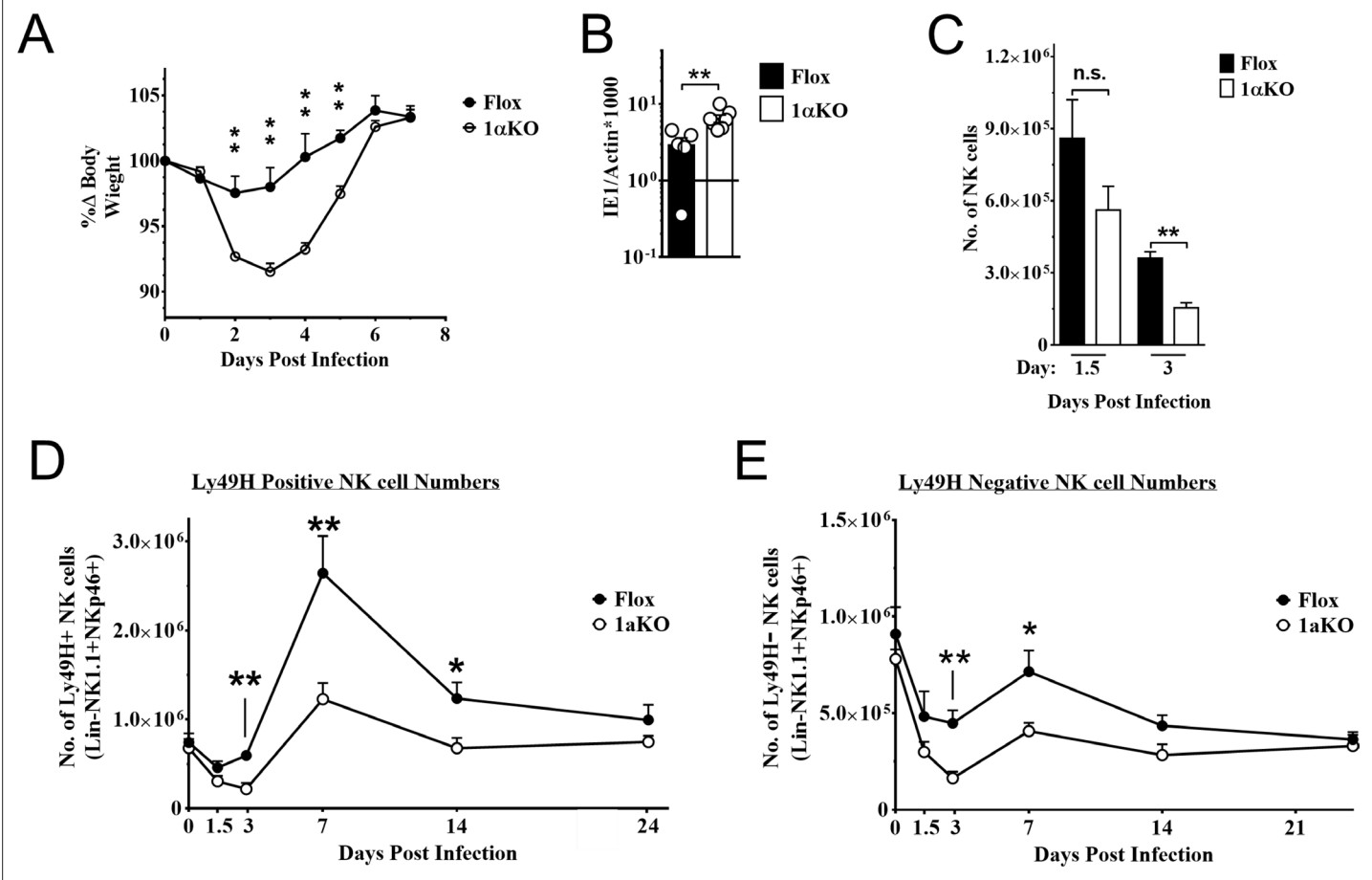

**Figure 2.** Natural killer (NK) cells require hypoxia-inducible factor-1α (HIF1α) for an optimal response to virus infection. (**A**) 1αKO or FL control mice were infected with 50 K plaque forming unit (pfu) then monitored for weight loss over 7 days. Each data represents four mice from two independent experiments. (**B**) Splenic viral load in 1αKO or FL control mice were infected with 50 K pfu then spleens harvested day 5 post-infection (pi). Data are pooled from four biologically independent experiments with a total of 5–7 mice in the indicated groups. (**C**) Number of bulk NK cells was quantified at days 1.5 and 3 pi from 1αKO or FL control mice. Data are from 2–3 independent experiments with 4–6 mice per group. (**D**) 1αKO or FL control mice were infected with 50 K pfu and Ly49H+ NK cell expansion was determined at days 1.5, 3, 7, 14, and 24 pi. Data are from three independent experiments with 4–9 mice per group. (**E**) Quantification of 1αKO or Flox Ly49H- NK cells at days 1.5, 3, 7, 14, and 24 post murine cytomegalovirus (MCMV) infection. Data are from 3–4 independent experiments with 3–4 mice per group. Data depict mean ± SEM, with each data set containing data indicated number of mice per group from independent experiments. Unpaired t-test was performed on (**A**–**D**). Statistical significance indicated by n.s., no significant difference; *p<0.05; **p<0.01; ***p<0.001; ****p<0.0001.

The online version of this article includes the following figure supplement(s) for figure 2:

**Figure supplement 1.** Normal development in hypoxia-inducible factor-1α (HIF1α) KO natural killer (NK) cells.

7 pi but not at other time points (**Figure 2E**). Thus, during MCMV infection, NK cells require HIF1α to protect against morbidity, control viral load, sustain NK cell numbers, and support Ly49H+ NK cell expansion.

## HIF1α is dispensable for NK cell effector function

To determine how HIF1α is required by NK cells to provide immunity during MCMV infection, we assessed if HIF1α contributed to NK cell activation and effector function. Surprisingly, we found that 1αKO NK cells at day 1.5 pi had significant increases in CD69 expression, and IFNγ production with decreased KLRG1 expression as compared to uninfected 1αKO NK cells (**Figure 3A**). While these changes were subtly more than FL NK cells, such changes were not readily apparent at day 3 pi (**Figure 3B**, **Figure 3—figure supplement 1A**). When we compared activation and effector markers between Ly49H+ and Ly49H- NK cells, we found no significant differences (**Figure 3—figure**

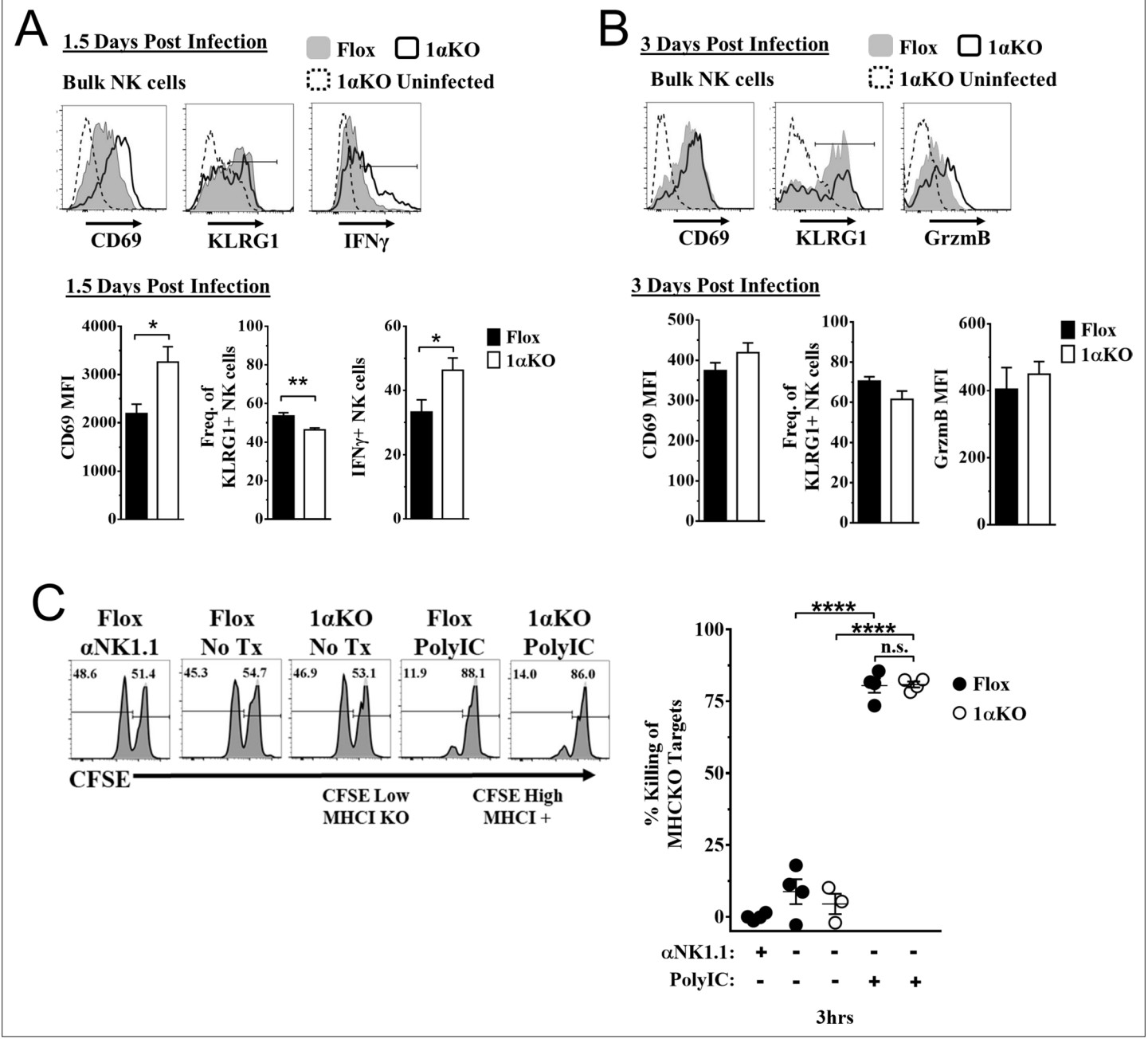

**Figure 3.** Hypoxia-inducible factor-1α (HIF1α) is dispensable for natural killer (NK) cell effector function. (**A**) Representative histograms (top) and quantification (bottom) of CD69 MFI on NK cells and frequency of KLRG1+ and IFNγ+ NK cells at day 1.5 post-infection (pi) of 1αKO or FL control mice. Data are from 2–3 independent experiments with 4–6 mice per group. (**B**) Representative histograms (top) and quantification (bottom) of CD69 and Granzyme B MFI (GrzmB) of NK cells, and frequency of KLRG1+ NK cells at day 3 pi. Data are from two independent experiments with four mice per group. (**C**) 1αKO or FL control mice were injected with PBS or Poly (**I:C**) then 3 days later 20 × 10^6 splenocytes at 1:1 ratio of CFSE-labeled MHCI-sufficient (CFSE^High) and -deficient (CFSE^Low) were intravenously transferred into mice. MHC-deficient splenocyte elimination was measured 3 hr post-transfer by flow cytometry. Data are from three independent experiments with 3–4 mice per group. Data depict mean ± SEM, with each data set containing data indicated number of mice per group from independent experiments. Unpaired t-test was performed on (**A**, **B**) or one-way ANOVA (**C**). Statistical significance indicated by n.s., no significant difference; *p<0.05; **p<0.01; ***p<0.001; ****p<0.0001.

The online version of this article includes the following figure supplement(s) for figure 3:

**Figure supplement 1.** Ly49H + and Ly49H- hypoxia-inducible factor-1α (HIF1α) KO natural killer (NK) cells have similar activation compared to Flox controls.

*supplement 1B*, *Figure 3—figure supplement 1C*). Thus, HIF1α is not required for general activation of NK cells during MCMV infection and may actually restrict NK activation early in infection.

To directly assess HIF1α-dependent NK cytolytic capacity in vivo without the confounding issue of decreased NK cell numbers during MCMV infection in 1αKO mice, we performed missing-self rejection assays in uninfected mice after TLR stimulation with poly-I:C. Both cohorts of mice were equally efficient killers of MHCI-deficient targets (*Figure 3C*). Thus, in vivo cytotoxic killing function is intact in HIF1αKO NK cells, suggesting that the impaired NK cell immunity to MCMV is not related to inability to kill target cells.

## HIF1αKO NK cells divide normally but numbers are reduced

Another possible explanation for the impaired NK cell immunity to MCMV observed in 1αKO mice could be a defect in NK cell proliferation. However, we observed a modest increase in the expression of proliferation marker Ki67 in bulk NK cells at day 1.5 pi and higher expression at day 3 pi that were comparable between infected 1αKO and control FL mice (*Figure 4A*). Similar results were seen with BrdU incorporation at day 3 (*Figure 4—figure supplement 1A*), confirming that HIF1α is not required during the non-specific phase of NK cell proliferation that occurs early after MCMV infection (*Dokun et al., 2001*).

At later time points, Ly49H+ NK cells demonstrate selective proliferation, resulting in expansion (*Dokun et al., 2001*; *Sun et al., 2009*). Since Ly49H expansion was reduced in 1αKO mice (*Figure 2D*), we specifically assessed proliferation of Ly49H+ NK cells. Surprisingly, Ki67 expression on Ly49H+ cells in 1αKO mice was comparable to FL controls (*Figure 4B*). Furthermore, the frequencies of Ly49H+ NK cells in both groups of mice were comparable at day 3 pi (*Figure 4—figure supplement 1B*). At the peak of Ly49H+ NK cell expansion at day 7 pi, there were again no differences in frequencies or Ki67 expression in 1αKO and FL controls (*Figure 4—figure supplement 1C*, *Figure 4—figure supplement 1D*). Thus, during MCMV infection, selective Ly49H+ NK cell proliferation is also normal in NK cells lacking HIF1α.

To extend our analysis of NK cell proliferation in vivo in the absence of HIF1α, we utilized a non-inflammatory in vivo model for homeostatic NK cell proliferation by adoptively transferring 1αKO or FL splenocytes into lymphopenic Rag-γRc mice. CFSE-labeled, HIF1α-sufficient and -deficient splenic NK cells analyzed at day 4 post transfer showed no significant differences in cell division (*Figure 4C*) but the frequency of 1αKO NK cells was reduced compared to FL NK cells (*Figure 4D*). At 2 weeks post transfer, reduced numbers of 1αKO NK cells were observed as compared to FL controls (*Figure 4E*) while all other immune cells showed comparable or even increased engraftment when derived from 1αKO mice (*Figure 4—figure supplement 1E*). Thus, HIF1α is dispensable for homeostatic proliferation in NK cells in vivo though ultimate expansion was affected.

Since cytokines such as IL-2 can induce NK cell proliferation and expansion, we tested its in vitro effects on HIF1α-deficient NK cells. Indeed, when enriched CFSE-labeled splenic 1αKO and FL NK cells were in vitro stimulated with different concentrations of IL-2, we found comparable cell division (*Figure 4F*). Interestingly, however, when NK cells were quantified after 6 days of culture, 1αKO NK cell numbers were significantly reduced compared to FL NK cells, though these differences were less evident at lower IL-2 concentrations (*Figure 4G*). Taken together, these data indicate that HIF1α controls NK cell numbers after proliferation though proliferation itself is HIF1α-independent.

## HIF1αKO NK cells are predisposed to apoptosis

Since 1αKO NK cells showed normal proliferation but did not expand, we hypothesized that HIF1α plays an unexpected role in NK cell survival. Since survival is mediated through signaling and inter-actions of pro-survival (Mcl-1, Bcl2) and pro-apoptotic proteins (Bim, cytochrome c), we examined if these factors were associated with HIF1α. At 3 days after stimulation, IL-2-activated 1αKO NK cells showed significant increases in transcripts for pro-apoptotic *Bim*, and *cytochrome c*, as compared to FL controls (*Figure 5A*), which was before 1αKO NK cells had significantly reduced numbers (*Figure 5—figure supplement 1A*). Furthermore, Bim and cytochrome c protein expression was significantly increased while protein expression of pro-survival Bcl-2 showed a significant decrease (*Figure 5B*). However, expression of a known HIF1α pro-survival gene target, Mcl-1 (*Carrington et al., 2017*; *Liu et al., 2006*; *Piret et al., 2005*), was not significantly different in 1αKO NK cells (*Figure 5—figure supplement 1B*, *Figure 5—figure supplement 1C*). Thus, HIF1α plays an unexpected role in survival

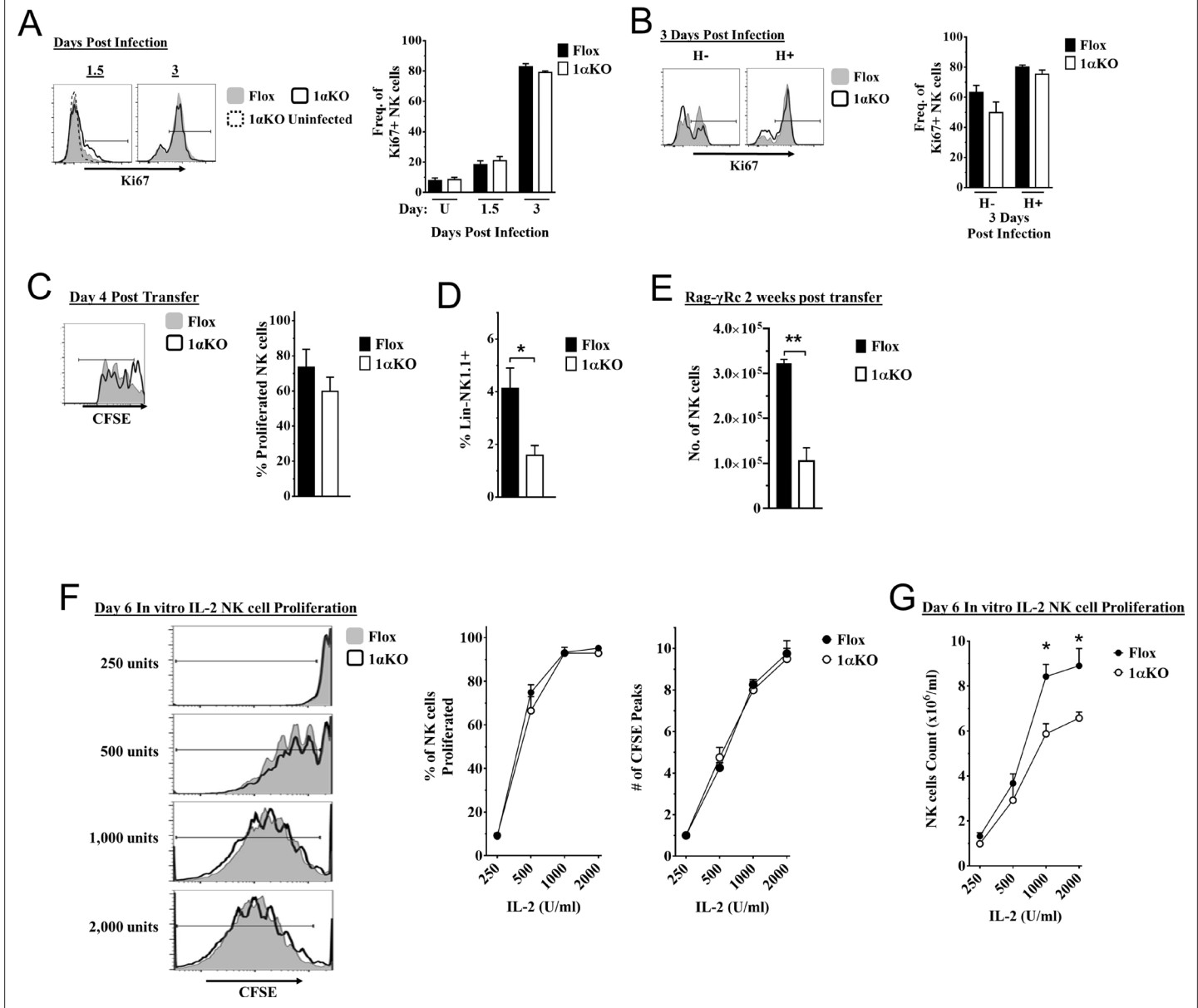

**Figure 4.** Cell division in hypoxia-inducible factor-1α (HIF1α) KO natural killer (NK) cells is normal but numbers are reduced. (**A, B**) 1αKO or FL control mice infected with murine cytomegalovirus (MCMV) received a dose of 50 K plaque forming unit (pfu) and analyzed as indicated. (**A**) Ki67 expression with representative histograms (left) and frequency quantification (right) in 1αKO or FL control bulk NK cells at day 1.5 and day 3 post-infection (pi). Data are from three independent experiments with 5–7 mice per group. (**B**) Expression of Ki67 in Ly49H negative and positive NK cells shown in histogram (left) and frequencies quantified (right). Data are from two independent experiments with three mice per group. (**C, D**) Rag-γRc mice analyzed for CFSE + splenic NK cells from 1αKO or FL control mice 4 days post transfer, showing representative histogram (**C**, left) and quantification of total proliferation (**C**, right), and frequencies (**D**). Data are from two independent experiments with four mice per group. (**E**) Rag-γRc mice analyzed on day 14 since post splenocyte transfer for numbers of NK cells in the spleen from 1αKO or FL control mice. Data are from two independent experiments with three mice per group. (**F, G**) In vitro proliferation assays were done using human recombinant IL-2 at different concentrations and time points indicated below. (**F**) Representative histograms (left), quantification of total proliferated cells (middle), and number of CFSE peaks (right) of CFSE labeled NK cells from 1αKO or FL control mice stimulated with different concentrations of IL-2 for 6 days. (**G**) Numbers of NK cells were counted from (**F**) and graphed in cells per 1 ml. Data are from two independent experiments with four mice per group. Data depict mean ± SEM, with each data set containing data indicated number of mice per group from independent experiments. Unpaired t-test was performed on (**A–G**). Statistical significance indicated by n.s., no significant difference; *$p < 0.05$; **$p < 0.01$; ***$p < 0.001$; ****$p < 0.0001$.

The online version of this article includes the following figure supplement(s) for figure 4:

**Figure supplement 1.** Proliferation is normal in hypoxia-inducible factor-1α (HIF1α) natural killer (NK) cells.

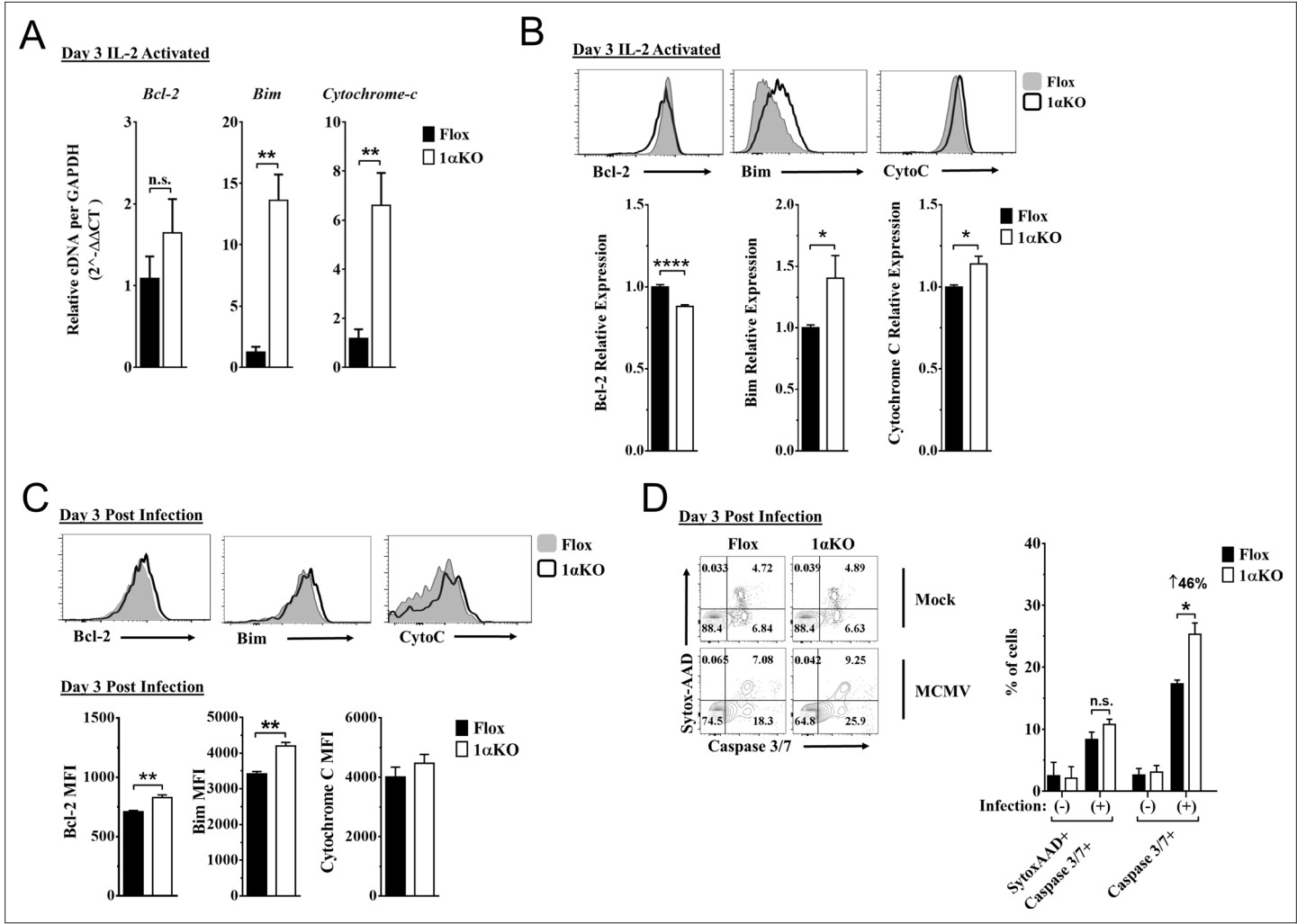

**Figure 5.** Hypoxia-inducible factor-1α (HIF1α) KO natural killer (NK) cells are predisposed to apoptosis. (**A**) mRNA transcripts for levels of apoptosis associated genes Bcl-2, Bim, and cytochrome c was measured in 1αKO or FL control NK cells at day 3 post IL-2 stimulation. Data are from two independent experiments with four mice per group. (**B**) Representative histograms (top) and quantification (bottom) of pro-survival relative protein expression of Bcl-2, Bim, and cytochrome c (Cyto-C) in NK cells 3 days post IL-2 stimulation. Data are from 2–5 independent experiments with 4–11 mice per group. (**C**) Representative histograms (top) and quantification (bottom) of Bcl-2 MFI, Bim MFI, and cytochrome c (Cyto-C) MFI in NK cells day 3 post murine cytomegalovirus (MCMV) infection. Data are from 2–3 independent experiments with 4–6 mice per group. (**D**) Representative contour plots (left) and frequency quantification (right) of necrotic cells (Caspase 3/7+ Sytox AAD+), and apoptotic cells (Caspase 3/7+ Sytox AAD-). Data are from two independent experiments with three mice per group. Data depict mean ± SEM, with each data set containing data indicated number of mice per group from independent experiments. Unpaired t-test was performed on (**A**–**D**). Statistical significance indicated by n.s., no significant difference; *p<0.05; **p<0.01; ***p<0.001; ****p<0.0001.

The online version of this article includes the following figure supplement(s) for figure 5:

**Figure supplement 1.** Hypoxia-inducible factor-1α (HIF1α) KO natural killer (NK) cells are predisposed to cell death.

gene expression rather than proliferation in IL-2-stimulated NK cells in vitro, suggesting that this effect might account for decreased accumulation of Ly49H+ NK cells during MCMV infection in vivo, despite normal proliferation (***Figures 2D, 4A and B***).

Indeed, during MCMV infection in vivo, 1αKO NK cells showed elevated Bim expression at day 1.5 and day 3 pi (***Figure 5—figure supplement 1D*** and ***Figure 5C***, respectively), consistent with when HIF1α expression was highest (day 3 pi, ***Figure 1F***) and the number of NK cells began to significantly decline (***Figure 2C–E***). When we compared the ratio of Bim to Bcl-2 between Flox and 1αKO NK cells, we found a trend for an increased ratio in 1αKO NK cells at the protein level (***Figure 5—figure supplement 1E***). We next sought to determine if there was increased cell death by staining for Caspase 3/7 and Sytox-AAD for apoptotic cells (single positive Caspase 3/7) and necrotic cells (double positive).

At day 3 pi, 1αKO NK cells displayed increased apoptotic activity (46% increase compared to FL controls) when examined directly ex vivo with no difference in necrotic cells (*Figure 5D*). HIF1α regulates the autophagy gene BNIP3, raising the possibility that abnormal autophagy could account for our findings. However, no significant differences in the autophagy markers p62 and CytoID were detected between 1αKO and FL control NK cells at day 3 pi (*Figure 5—figure supplement 1F*). Taken together, these data show that HIF1α protein plays an important role in regulating survival, not proliferation, of NK cells during MCMV infection.

## Impaired glycolytic activity contributes to impaired fitness in HIF1αKO NK cells

Our data indicate that 1αKO NK cells do not have impaired cell division but rather show a predisposition for apoptosis that results in reduced numbers compared to FL controls during cytokine activation. We verified that HIF1α-deficient NK cells manifest increased OXPHOS (*Figure 6—figure supplement 1*) that was previously reported but whose basis was not further evaluated (*Ni et al., 2020*). Since these findings suggest that NK cell metabolism may be dysregulated in the absence of HIF1α, we ascertained if HIF1α-associated glycolytic genes were involved despite being considered dispensable (*Loftus et al., 2018*; *O'Brien and Finlay, 2019*). First, we determined if these metabolic changes could be due to glucose uptake defects since HIF1α affects glucose transport genes (*Chen et al., 2001*; *Huang, 2012*). However, despite verification of absence of HIF1α in 1αKO NK cells by direct measurement of *Hif1a* transcripts at 3 days post IL-2 stimulation (*Figure 6A*), we found that glucose uptake was normal as uptake of the 2-NDBG fluorescent glucose analog showed no defect (*Figure 6B*). By contrast, the expression of the HIF1α glycolytic target genes, hypoxanthine-guanine phosphoribosyl transferase (*Hrpt*), glucose transporter type 1 (*Glut1*), pyruvate kinase M2 (*Pkm2*), hexokinase 2 (*Hk2*), glucose-6-phosphate isomerase 1 (*Gpi1*), lactate dehydrogenase A (*Ldha*), and phosphoinositide-dependent kinase 1 (*Pdk1*), was markedly reduced in 1αKO NK cells (*Figure 6C*). Furthermore, we found an impaired utilization of glucose and production of lactate when we measured glucose metabolism using liquid chromatography mass spectrometry (LCMS) (*Figure 6D*). However, no defects were found in glycolysis-independent metabolism, that is glutamine/glutamate pathway. Thus, HIF1α affects metabolism of NK cells by supporting glycolysis.

## Discussion

Here we define a previously undescribed role for HIF1α in supporting NK cell-mediated protection against MCMV infection. In the selective absence of HIF1α in NK cells, we found increased morbidity and viral burden. However, we found that HIF1αKO NK cells displayed normal activation and proliferation in response to MCMV but they did not normally expand in number. This defect was evident in another in vivo response, that is homeostatic proliferation in lymphopenic mice, as well as with in vitro IL2 stimulation where there was normal cell division but impaired expansion. The 1αKO NK cells had an increase in apoptosis as a result of decreased expression of glycolytic genes and abnormal glucose metabolism that have been previously associated with apoptosis (*Dengler et al., 2014*; *El Mjiyad et al., 2011*; *Greer et al., 2012*; *MacFarlane et al., 2012*). Thus, HIF1α supports survival in NK cells through providing optimal glucose metabolism during pathogen infection.

While HIF1α protein function in NK cells during pathogen infection had not been detailed previously, it was recently explored in tumor-infiltrating NK cells in two studies with comparable results attributed to different mechanisms (*Krzywinska et al., 2017*; *Ni et al., 2020*). In mice with HIF1α-deficient NK cells, tumor control was enhanced either directly by increased IL-18 signaling that improved NK cell effector responses or indirectly through an absence of tumor-infiltrating NK cells, which altered the VEGF/VEGFR axis affecting productive angiogenesis. In contrast, our findings were the polar opposite to these reports in that mice with HIF1α-deficient NK cells showed reduced control of viral infection without alteration in NK cell activation or killing ability. While HIF1α-deficient NK cell numbers were normal in the naïve spleen, after MCMV infection there was a significant decrease in numbers with impaired expansion of Ly49H+ cells that we attribute to apoptosis. While we used the same strain of HIF1α-FL mice, it is possible that there are subtle genetic differences due to backcrossing from the 129 genetic background to C57BL/6, as is the case with all backcrossed mice. In addition, another potential source of these disparities is that our Ncr1-cre mice differ than those

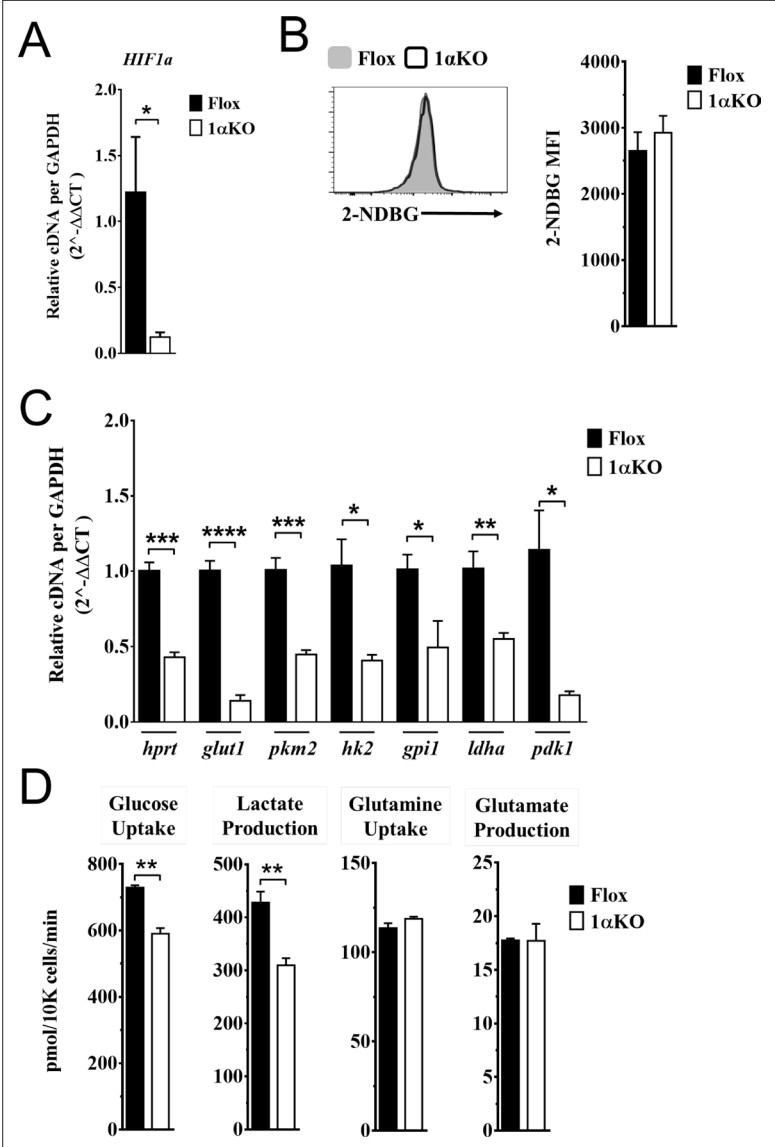

**Figure 6.** Impaired glycolytic activity contributes to impaired survival of hypoxia-inducible factor-1α (HIF1α) KO natural killer (NK) cells. Enriched NK cells from 1αKO or FL control mice were IL-2 treated for 3 days then collected and analyzed as follows. (**A**) mRNA transcript levels of *HIF1α* for both cohort of mice measured with qRT-PCR. Data are from two independent experiments with four mice per group. (**B**) Flow cytometry analysis of 2-NDBG MFI expression as a measure of glucose intake. Data are from three independent experiments with six mice per group. (**C**) mRNA transcripts levels for glycolysis genes *glut1*, *gpi1*, *hk2*, *hprt*, *ldha*, *pdk1*, and *pkm2* were measured with qRT-PCR. Data are from two independent experiments with four mice per group. (**D**) Media was collected from day 3 IL-2-activated NK cell cultures and using LC/MS, uptake of glucose (first panel) and glutamine (third panel), and production of lactate (second panel) and glutamate (fourth panel) was measured. Data are representative of two independent experiments with three mice per group. Data depict mean ± SEM, with each data set containing data indicated number of mice per group from independent experiments. Unpaired t-test was performed on (**A–D**). Statistical significance indicated by n.s., no significant difference; *p<0.05; **p<0.01; ***p<0.001; ****p<0.0001.

The online version of this article includes the following figure supplement(s) for figure 6:

**Figure supplement 1.** Hypoxia-inducible factor-1α (HIF1α) KO natural killer (NK) cells have dysregulated metabolism.

used in the aforementioned studies. Nonetheless, the function of HIF1α in NK cells appears markedly different in pathogen responses as compared to NK cell responses to tumors.

Here we provide evidence that the absence of HIF1α affects survival in NK cells during a pathogen infection. HIF1α senses intracellular changes and supports transcription of genes related to survival in several cellular pathways (*Dengler et al., 2014*; *Daskalaki et al., 2018*; *Krzywinska and Stockmann, 2018*; *Majmundar et al., 2010*). In the apoptosis pathway, constitutively expressed pro- and anti-apoptotic proteins antagonize each other to sustain viability in normal healthy cells, and when these proteins are unbalanced due to external or internal stimuli, pro-apoptotic proteins are preferentially activated and expressed, resulting in the initiation of programmed cell death via caspase activity (*El Mjiyad et al., 2011*; *Iurlaro and Muñoz-Pinedo, 2016*; *Thomas et al., 2010*). In tumor models, HIF1α can directly impact apoptosis by upregulating the pro-survival gene myeloid cell leukemia sequence-1 (Mcl-1) (*Carrington et al., 2017*; *Liu et al., 2006*; *Piret et al., 2005*). Mcl-1 was found to be highly expressed in normal NK cells, and genetic ablation of Mcl-1 in NK cells resulted in a dramatic loss of mature NK cells at baseline that spared immature populations (*Sathe et al., 2014*). This absence of NK cells impaired control of tumors as well as protection against lethal sepsis. Here Mcl-1 is unlikely to contribute to the HIF1α-deficient NK cell phenotype because we found no difference in transcript or protein levels of Mcl-1 in 1αKO compared to FL control NK cells, and there were no baseline differences in NK cell numbers. Another potential survival pathway regulated by HIF1α in tumor cells is through the increased expression of BCL2/adenovirus E1B 19 kDa protein-interacting protein 3 (BNIP3), a component of autophagosomes that complexes with p62 LC3 complex proteins to initiate a survival mechanism of self-renewal. In MCMV infection, O'Sullivan et al. reported that BNIP3 supported memory formation in NK cells (*O'Sullivan et al., 2015*). Wild-type (WT) Ly49H + NK cells were transferred into Ly49H-deficient mice; upon infection, memory NK cells displayed levels of BNIP3 transcripts comparable to resting NK cells. However, in a competition experiment, *Bnip3⁻/⁻* NK cells showed impaired survival during the contraction phase as compared to WT NK cells. The markers CytoID and LC3 that indicated active autophagy were comparable between resting and memory WT NK cells but autophagy was impaired in memory *Bnip3⁻/⁻* NK cells, which displayed higher CytoID expression. Interestingly, BNIP3 transcripts were reduced during the effector phase of the response, but this was not explored and the interplay between BNIP3 and HIF1α was not determined. In our studies, there were no differences in CytoID and p62 at 3 days post MCMV infection in 1αKO and FL NK cells, suggesting that this pro-survival axis is unaffected in the absence of HIF1α. Finally, when survival proteins are unaltered, a mechanism of subverting survival is the overexpression of pro-apoptotic proteins. Multiple pathways converge on the potent inducer of apoptosis, Bim, which stimulates caspase activity (*El Mjiyad et al., 2011*; *Iurlaro and Muñoz-Pinedo, 2016*; *Thomas et al., 2010*). Bim regulates NK cell survival and restrains the expansion of memory cells during MCMV infection (*Huntington et al., 2007*; *Min-Oo et al., 2014*). Bim can potentiate apoptosis even in the presence of sufficient pro-survival signals in NK cells, but the interplay between HIF1α and Bim was not previously explored (*Goh et al., 2020*). In our experiments, HIF1α-deficient NK cells had elevated levels of Bim as compared to controls during in vitro proliferation and MCMV infection. Furthermore, HIF1α-deficient NK cells showed a marked increase in caspase activity, a direct measurement of apoptosis, as compared to Flox controls at 3 days pi. Thus, in our studies we found that activated HIF1α-deficient NK cells upregulate the key pro-apoptotic protein Bim and its downstream targets during in vitro proliferative and pathogen infection responses to promote cell death while there was little evidence implicating other survival pathways.

Our studies indicate that HIF1α-deficient NK cells have impaired metabolism of glucose that provides fuel for glycolysis, critical for cellular survival (*Dengler et al., 2014*; *El Mjiyad et al., 2011*; *Greer et al., 2012*; *MacFarlane et al., 2012*; *Majmundar et al., 2010*; *Donnelly and Finlay, 2015*). The importance of glycolysis in NK cells has been previously studied using glycolytic inhibitors. *Keppel et al., 2015*; *Mah et al., 2017* inhibited glucose metabolism during acute activation, and during MCMV infection that resulted in a reduction of IFNγ responses, and impaired proliferation and protective immunity, respectively. However, data from our studies here and others (*Ni et al., 2020*; *Chang et al., 2013*) suggest that HIF1α-associated glucose metabolism is dispensable for effective IFNγ responses in NK cells. Moreover, *Loftus et al., 2018* reported that HIF1α was not required for NK cell activation, transcription of glycolytic genes, and glucose metabolism. Further, it was determined that cMyc was the potent regulator of glucose metabolism, leading to the conclusion that HIF1α was

dispensable in NK cells for glucose metabolism (*O'Brien and Finlay, 2019*). One potential important caveat for these studies is that NK cells were expanded for 6 days in IL-15 in vitro then stimulated with IL-2 and IL-12 for activation (*Loftus et al., 2018*). In another in vitro study, enriched HIF1α-deficient NK cells were IL-2 stimulated for 7 days in hypoxic conditions then stimulated for 4–6 hr with IL-12 and IL-18 (*Ni et al., 2020*); deletion of HIF1α in NK cells resulted in higher OXPHOS, presumably to compensate impaired glucose metabolism. However, in this same experiment, glucose metabolism was comparable in both HIF1α-deficient and -sufficient NK cells. In our studies here, we measured metabolism directly ex vivo and found increases in both glucose metabolism and OXPHOS in resting HIF1αKO NK cells compared to Flox controls. Thus, prior results were skewed by culture conditions that likely do not reliably represent in vivo effects of HIF1α on glucose metabolism in NK cells.

In further contrast to the aforementioned studies, we found that in the absence of HIF1α in NK cells, IL-2 stimulation resulted in a reduced glycolytic transcriptome and a subsequent reduction in glucose metabolism. These experiments coincide with other IL-2 stimulation assays we performed where pro-apoptotic Bim was upregulated at the gene and protein level. Importantly, reduced glucose metabolism has been directly associated with apoptosis via Bim-caspase-3 pathway (*El Mjiyad et al., 2011*; *MacFarlane et al., 2012*; *Iurlaro and Muñoz-Pinedo, 2016*; *Shin et al., 2015*). Briefly, we speculate that poor glucose metabolism in HIF1α-deficient NK cells imitates glucose deprivation over the course of MCMV infection. Studies have shown that during glucose deprivation upregulation and activation of Bim can be induced by nutrient sensors AMPK and Akt, and the endoplasmic reticulum stress pathway. Normally, MCMV infection causes the production of pro-inflammatory cytokines that activate and proliferate NK cells (*Arase et al., 2002*; *Biron and Tarrio, 2015*; *Mitrović et al., 2012*; *Pyzik and Vidal, 2009*; *Smith et al., 2002*). To sustain ATP demands for activation and products for biosynthesis, we hypothesize that NK cells metabolically adapt through cellular respiration (*Graham and Presnell, 2017*; *Kaelin, 2018*; *Schofield and Ratcliffe, 2004*; *Semenza, 2014*). In turn, oxygen consumption causes stabilization of HIF1α and its translocation into the nucleus (*Dengler et al., 2014*; *Schödel et al., 2011*). In summary, expression of HIF1α gene targets is necessary to provide sufficient glucose metabolism to support survival by repressing Bim protein expression, thereby sustaining adequate numbers of NK cells for protective immunity.

Targeting HIF1α using small molecule inhibitors has drawn considerable interest over recent years for the treatment of multiple pathologies. However, HIF1α is ubiquitously expressed and has pleiotropic properties through targeting of multiple genes, as illustrated here and elsewhere (*Dengler et al., 2014*; *Schödel et al., 2011*). Therefore, special care should be taken when considering the use of HIF1α inhibitors as they may compromise NK cell function under certain conditions such as pathogen infection.

# Materials and methods

## Key resources table

| Reagent type (species) or resource | Designation | Source or reference | Identifiers | Additional information |
|---|---|---|---|---|
| Strain, strain background (*Mus musculus*) | β2m KO; B6.129P2-B2mtm1Unc/J | Jackson Laboratories | IMSR Cat# JAX:002070, RRID:IMSR_JAX:002070 | |
| Strain, strain background (*Mus musculus*) | HIF1a Flox; B6.129-Hif1atm3Rsjo/J | Jackson Laboratories | IMSR Cat# JAX:007561, RRID:IMSR_JAX:007561 | |
| Strain, strain background (*Mus musculus*) | NKp46iCre; Ncr1tm1.1(icre)Viv/J | Eric Vivier | MGI Cat# 5309017, RRID:MGI:5309017 | |
| Strain, strain background (*Mus musculus*) | Wild-type; C57BL/6NCr | Charles River | MGI Cat# 5657970, RRID:MGI:5657970 | |
| Strain, strain background (murine cytomegalovirus) | MCMV | *Cheng et al., 2010* | | |

*Continued on next page*

*Continued*

| Reagent type (species) or resource | Designation | Source or reference | Identifiers | Additional information |
|---|---|---|---|---|
| Antibody | Anti-SQSTM1/p62 Alexa Fluor 647 (rabbit monoclonal) | Abcam | Cat# ab194721 | Flow (1:500) |
| Antibody | Anti-CD49a Brilliant Violet 421 (Armenian hamster monoclonal) | BD Biosciences | Cat# 740046, RRID:AB_2739815 | Flow (1:100) |
| Antibody | Anti-Bcl-2 Alexa Fluor 488 (mouse monoclonal) | BioLegend | Cat# 633506, RRID:AB_2028390 | Flow (1:100) |
| Antibody | Anti-Cytochrome c Alexa Fluor 647 (mouse monoclonal) | BioLegend | Cat# 612310, RRID:AB_2565241 | Flow (1:100) |
| Antibody | Anti-DNAM-1 PE (rat monoclonal) | BioLegend | Cat# 132006, RRID:AB_1279173 | Flow (1:100) |
| Antibody | Anti-HIF1a PE (mouse monoclonal) | BioLegend | Cat# 359704, RRID:AB_2562423 | Flow (5:100) |
| Antibody | Anti-Ly49A PE (rat monoclonal) | BioLegend | Cat# 116808, RRID:AB_313759 | Flow (1:200) |
| Antibody | Anti-NK1.1 BV650 (mouse monoclonal) | BioLegend | Cat# 108736, RRID:AB_2563159 | Flow (1:100) |
| Antibody | Anti-Mcl-1 PE (rabbit monoclonal) | Cell Signaling Technology | Cat# 65617, RRID:AB_2799688 | Flow (1:50) |
| Antibody | APC-AffiniPure F(ab')2 Fragment Donkey Anti-Goat IgG (H + L) (min X Ck,GP,Sy Hms,Hrs,Hu,Ms,Rb,Rat Sr Prot) antibody (polyclonal) | Jackson ImmunoResearch Labs | Cat# 705-136-147, RRID:AB_2340407 | Flow (1:300) |
| Antibody | Anti-Ly49H FITC (mouse monoclonal) | Made in-house FACS | | Flow (1:100) |
| Antibody | Anti-Bim (rabbit monoclonal) | Thermo Fisher Scientific | Cat# MA5-14848, RRID:AB_10981505 | Flow (1:200) |
| Antibody | Anti-CD3e AF700 (Syrian hamster monoclonal) | Thermo Fisher Scientific | Cat# 56-0033-82, RRID:AB_837094 | Flow (2:100) |
| Antibody | Anti-CD19 Alexa Fluor 700 (rat monoclonal) | Thermo Fisher Scientific | Cat# 56-0193-80, RRID:AB_837082 | Flow (1:100) |
| Antibody | Anti-CD11b eFluor 450 (rat monoclonal) | Thermo Fisher Scientific | Cat# 48-0112-82, RRID:AB_1582236 | Flow (1:100) |
| Antibody | Anti-CD27 PE-Cy7 (Armenian hamster monoclonal) | Thermo Fisher Scientific | Cat# 25-0271-82, RRID:AB_1724035 | Flow (1:100) |
| Antibody | Anti-CD49b APC (rat monoclonal) | Thermo Fisher Scientific | Cat# 17-5971-82, RRID:AB_469485 | Flow (1:100) |
| Antibody | Anti-CD69 PE-Cy7 (Armenian hamster monoclonal) | Thermo Fisher Scientific | Cat# 25-0691-82, RRID:AB_469637 | Flow (1:100) |
| Antibody | Anti-Granzyme B APC (mouse monoclonal) | Thermo Fisher Scientific | Cat# MHGB05, RRID:AB_10373420 | Flow (5:100) |
| Antibody | Anti-IFNg eFluor450 (rat monoclonal) | Thermo Fisher Scientific | Cat# 48-7311-82,, RRID:AB_1834366 | Flow (1:100) |
| Antibody | Anti-Ki-67 eFluor 450 (rat monoclonal) | Thermo Fisher Scientific | Cat# 48-5698-80, RRID:AB_1115115 | Flow (1:100) |

*Continued on next page*

*Continued*

| Reagent type (species) or resource | Designation | Source or reference | Identifiers | Additional information |
|---|---|---|---|---|
| Antibody | Anti-KLRG1 APC (Syrian hamster monoclonal) | Thermo Fisher Scientific | Cat# 17-5893-82, RRID:AB_469469 | Flow (1:100) |
| Antibody | Anti-Ly49G PE (mouse monoclonal) | Thermo Fisher Scientific | Cat# 12-5885-82, RRID:AB_466005 | Flow (2:100) |
| Antibody | Anti-Ly49H APC (mouse monoclonal) | Thermo Fisher Scientific | Cat# 17-5886-82, RRID:AB_10598809 | Flow (1:100) |
| Antibody | Anti-Ly49I FITC (mouse monoclonal) | Thermo Fisher Scientific | Cat# 11-5895-85, RRID:AB_465302 | Flow (1:100) |
| Antibody | Anti-NKG2A PE (mouse monoclonal) | Thermo Fisher Scientific | Cat# 12-5897-82, RRID:AB_46602 | Flow (2:100) |
| Antibody | Anti-NKp46 PerCPeFluor710 (rat monoclonal) | Thermo Fisher Scientific | Cat#: 46-3351-82, RRID:AB_1834441 | Flow (1:100) |
| Other | Viability stain eFluor 506 | Thermo Fisher Scientific | Cat# 65-0866-14 | Flow (1:1000) |
| Other | Viability stain eFluor 780 | Thermo Fisher Scientific | Cat# 65-0865-14 | Flow (1:1000) |
| Sequence-based reagent | Actin b | IDT DNA | TAQman assay | Forward: 5'-AGCTCATTGTAGAAGGTGTGG-3'; reverse: 5'-GGTGGGAATGGGTCAGAAG-3'; probe: 5'-TTCAGGGTCAGGATACCTCT CTTGCT-3' |
| Sequence-based reagent | Bcl-2 | IDT DNA | PowerSYBR Green PCR Master Mix | Forward: 5'-CCTGTGGATGACTGAG TACCTG-3'; reverse; 5'-AGCCAGGAGAAATCAA ACAGAGG-3' |
| Sequence-based reagent | Bim | IDT DNA | PowerSYBR Green PCR Master Mix | Forward: 5'-GGAGATACGGATTGCA CAGGAG-3'; reverse: 5'-CTCCATACCAGA CGGAAGATAAAG-3' |
| Sequence-based reagent | Cytochrome c | IDT DNA | PowerSYBR Green PCR Master Mix | Forward: 5'-GAGGCAAGCATAAGAC TGGACC-3'; reverse: 5'-ACTCCATCAGGGTATC CTCTCC-3' |
| Sequence-based reagent | gapdh | IDT DNA | PowerSYBR Green PCR Master Mix | Forward: 5'-GTTGTCTCCTGCGACTTCA-3'; reverse: 5'-GGTGGTCCAGGGTTTCTTA-3' |
| Sequence-based reagent | glut1 | IDT DNA | PowerSYBR Green PCR Master Mix | Forward: 5'-AAGAAGCTGACGGGTCGCCT CATGC-3'; reverse: 5'-TGAGAGGGACCAGAGC GTGGTG-3' |
| Sequence-based reagent | gpi1 | IDT DNA | PowerSYBR Green PCR Master Mix | Forward: 5'-GTTGCCTGAAGAGGCCAGG-3'; reverse: 5'-GCTGTTGCTTGATGAAGCTGATC-3' |
| Sequence-based reagent | hif1a | IDT DNA | PowerSYBR Green PCR Master Mix | Forward: 5'-CATCAGTTGCCACTTCCCCA-3'; reverse: 5'-GGCATCCAGAAGTTTTCTCACAC-3' |
| Sequence-based reagent | hk2 | IDT DNA | PowerSYBR Green PCR Master Mix | Forward: 5'-GGAGAGCACGTGTGACGAC-3'; reverse: 5'-GATGCGACAGGCCACAGCA-3', |
| Sequence-based reagent | hprt | IDT DNA | PowerSYBR Green PCR Master Mix | Forward: 5'- CTGGTGAAAAGGACCT CTCGAAG-3'; reverse: 5'- CCAGTTTCACTA ATGACACAAACG-3' |
| Sequence-based reagent | ldha | IDT DNA | PowerSYBR Green PCR Master Mix | Forward: 5'-CACAAGCAGGTGGTGGACAG-3'; reverse: 5'-AACTGCAGCTCCTTCTGGATTC-3' |
| Sequence-based reagent | mcl-1 | IDT DNA | PowerSYBR Green PCR Master Mix | Forward: 5'-AGCTTCATCGAACCAT TAGCAGAA-3'; reverse: 5'-CCTTCTAGGTCC TGTACGTGGA-3' |
| Sequence-based reagent | mcmv ie1 | IDT DNA | TAQman assay | Forward: 5'-CCCTCTCCTAACTCTCCCTTT-3'; reverse: 5'-TGGTGCTCTTTTCCCGTG 3'; probe: 5'-TCTCTTGCCCCGTCCTGAAAACC-3' |

*Continued on next page*

*Continued*

| Reagent type (species) or resource | Designation | Source or reference | Identifiers | Additional information |
|---|---|---|---|---|
| Sequence-based reagent | pdk1 | IDT DNA | PowerSYBR Green PCR Master Mix | Forward: 5'-GATTCAGGTTCACGTCACGCT-3'; reverse: 5'-GACGGATTCTGTCGACAGAG-3' |
| Sequence-based reagent | pkm2 | IDT DNA | PowerSYBR Green PCR Master Mix | Forward: 5'-CAGGAGTGCTCACCAAGTGG-3'; reverse: 5'-CATCAAGGTACAGGCACTACAC-3' |
| Commercial assay or kit | XF Cell Mito Stress Test Kit | Agilent | Cat# 103015-100 | |
| Commercial assay or kit | CYTO-ID Autophagy detection kit | Enzo Life Sciences, Inc. | Cat# ENZ-51031-0050 | |
| Commercial assay or kit | Cell Event Caspase 3/7 Green Flow Cytometry Assay Kit | Thermo Fisher Scientific | Cat# C10423 | |
| Chemical compound, drug | Recombinant mouse IL-18 | MBL | Cat# B002-5 | |
| Chemical compound, drug | Recombinant murine IL-15 | Peptrotech | Cat# 210-15 | |
| Chemical compound, drug | Interferon-beta | PBL Assay Science | Cat# 12401-1, RRID:AB_605469 | |
| Chemical compound, drug | Percoll | Sigma | Cat# P1644 | |
| Software, algorithm | FlowJo 10 | Treestar | https://www.flowjo.com/; RRID:SCR_008520 | |
| Software, algorithm | Prism 6 | GraphPad | https://www.graphpad.com/; RRID:SCR_002798 | |

## Mice

Wild-type B6 mice (C57/BLNCR1) were purchased through Charles River Laboratories (Wilmington, MA). *HIF1α*-specific deletion in NK cells was achieved through crossing B6.129-*Hif1a<tm3Rsjo>*/J mice from The Jackson Laboratory (Bar Harbor, ME) with *Ncr1*[iCre] mice from Eric Vivier. Control *β2m* knockout mice (B6.129P2-*B2m<tm1Unc>*/J) were obtained from The Jackson Laboratory.

## NK cell culture and splenocyte transfer

Spleens from mice were collected and placed in complete RPMI (Gibco#11875 RPMI 1640, 10% fetal calf serum, 1% penicillin/streptomycin) and minced through 100 µm filter to create single-cell suspensions. Splenocytes were then RBC lysed with lysis buffer (1 M NH$_4$NaCl, 1 M HEPES) for 2 min and washed with complete RPMI. Splenocytes were counted and cultured at $10 \times 10^6$ cells per ml in complete RPMI, 10 µM non-essential amino acids, 0.57 µM 2-mercaptoethenol. When NK cell enrichment was used, NK cells were isolated from splenocytes using Stem Cell Technologies NK isolation kit (#19855) as described by the manufacturer. Cytokines were obtained from Peprotech: IL-15 (210-15), IL-18 (B002-5), Pestka Biomedical Laboratories assay science IFNβ (12401-1), and used as indicated in figure legends. Splenocyte cultures were incubated at 37 °C in a normal incubator (20 % O$_2$) or hypoxic chamber (1 % O$_2$). For intracellular cytokine measurements, NK cells were cultured for 20 hr then brefeldin A was added for the last 4 hr. Liver NK cells were enriched using 35 % Percoll in complete RPMI and centrifuged for 15 min at 25° C. Cells were subsequently RBC lysed for 2 min, washed once with complete RPMI, then stained for flow cytometry. Receptor stimulation was carried out by coating 96-well plates with 25 µg/ml of PK136 (NK1.1) or 0.5 µg/ml PMA (Sigma) and 4 µg/ml ionomycin (Sigma), then $2 \times 10^6$ splenocytes were added per well in the presence of brefeldin A (eBioscience) for 6 hr and analyzed. 24 hr cultures had no brefeldin A initially and 10 ng/ml of IL-15 added overnight, then fresh IL-15 and brefeldin A was added for the last 5 hr of culture. For homeostatic proliferation, $20 \times 10^6$ CFSE-labeled splenocytes from Flox or HIF1αKO mice were intravenously transferred into Rag-γRc mice. NK cell proliferation was measured from the spleen at 4 days post transfer and immune cell analysis on day 14 post transfer.

## Nutrient uptake analysis

For nutrient uptake analysis with LC/MS, enriched NK cells from FL and 1αKO mice were cultured in 96-well plate with media containing 2000 units of recombinant human IL-2. Media were refreshed on day 1. On day 3 after a further 48 hr incubation, the spent media were collected and analyzed. Known concentrations of U-$^{13}$C internal standards (glucose, lactate, glutamine, and glutamate; Cambridge Isotopes) were spiked into media samples before extraction. Extractions were performed in glass to avoid plastic contamination as previously reported (*Yao et al., 2016*). Media samples were measured by liquid chromatography coupled mass spectrometry (LC/MS) analysis, with a method previously described (*Yao et al., 2018*). Samples were run on a Luna aminopropyl column (3 μm, 100 mm × 2.0 mm I.D., Phenomenex) in negative mode with the following buffers and linear gradient: A = 95 % water, 5 % acetonitrile (ACN), 10 mM ammonium hydroxide, 10 mM ammonium acetate; B = 95 % ACN, 5 % water; 100% to 0% B from 0 to 45 min and 0 % B from 45 to 50 min; flow rate 200 μl/min. Mass spectrometry detection was carried out on a Thermo Scientific Q Exactive Plus coupled with ESI source. For each compound, the absolute concentrations were determined by calculating the ratio between the fully unlabeled peak from samples and the fully labeled peak from standards. The consumption rates were normalized by cell growth over the experimental time period (*Yao et al., 2019*).

## Seahorse studies

NK cells were enriched from FL and 1αKO mice and $4 \times 10^5$ cells were transferred into 96-well plate without any stimulation. Agilent Seahorse XF Cell Mito Stress test was performed on NK cells using oligomycin (10 μM), FCCP (10 μM), rotenone (10 μM) plus antimycin A (10 μM) over 2 hr. Oxygen consumption rate (OCR) and extracellular acidification rate (ECAR) was measured and analyzed using Agilent Seahorse XFe96 Analyzers.

## MCMV infection

MCMV stocks were derived from salivary gland homogenates and titered using 3T12 (ATCC CCL-164) fibroblast cell line. The complete genomic sequence for the strain used here was previously published by our lab (*Cheng et al., 2010*). Virus was inoculated intraperitoneally (IP) in a total volume of 200 μl PBS at indicated plaque forming unit (pfu) doses and mouse spleens were harvested at indicated time points for analysis. Morbidity was measured in weight loss everyday post-infection for 7 days. Viral load was measured as previously described (*Parikh et al., 2015*).

## In vivo cytotoxicity assay

Target splenocytes were isolated from β2m knockout mice and labeled with 2 μM CFSE (Life Technologies) and WT C57BL/6 controls were labeled with 10 μM CFSE. CFSE$^{low}$ and CFSE$^{high}$ target cells were mixed at a 1:1 ratio and $2 \times 10^6$ target cells were injected i.v. into FL or 1αKO mice that had received PBS or Poly:I-C inoculations 3 days prior. After 3 hr, splenocytes were harvested and stained. The ratio of CFSE$^{low}$ to CFSE$^{high}$ viable bulk CD45$^+$ splenocyte cells was determined by flow cytometry. Target cell rejection was calculated using the formula [(1− (Ratio (CFSE$^{low}$: CFSE$^{high}$) sample/Ratio (CFSE$^{low}$: CFSE$^{high}$) average of NK depleted)) × 100].

## Flow cytometry

After preparation, cells were stained in staining buffer (PBS, 0.2% fetal bovine serum, 0.01% sodium azide), and NK cells were identified as e780-CD45+ CD3-CD19-NK1.1+ NKp46+. Antibodies and reagents from eBioscience: anti-CD3ε (eBio500A2), anti-CD19 (eBio1D3), anti-NK1.1 (PK136), anti-NKp46 (29A1.4), anti-CD11b (M1/70), anti-IFN-γ (XMG1.2), anti-KLRG1(2F1), anti-NKG2A.B6 (16a11), anti-Ly49G2 (eBio4D11), anti-Ly49H (3D10), anti-Ly49I (YLI-90), anti-CD69 (H1.2F3), Fixable Viability Dye eFluor 780 (65-0865-14), anti-Ki67 (48-5698-82), Brefeldin A (00-4506-51), Transcription Factor Staining kit (00-5521); Invitrogen: Granzyme B (MHGB4); BD Pharmingen: anti-Ly49A (116805), BrdU (51-23614 L); BD Biosciences: BD Cytofix/Cytoperm kit (554714), anti-CD49a (Ha31/8); BioLegend: anti-NK1.1 (PK136), anti-HIF1α (546-16), anti-Bcl-2 (10C4), anti-cytochrome c (6H2,B4), anti-DNAM-1 (132006); Jackson ImmunoResearch: Alexa Fluor 647 F (ab')$_2$ Fragment Goat Anti-Rabbit IgG (111-606-046), RC fragment specific; Invitrogen: anti-Bim (K.912.7), CYTO-ID (ENZ-51031-0050); Abcam: SQSTM1/p62 antibody (ab194721); Cell Signaling: Mcl-1 (65617 S). Liver NK cells were

identified as e780-CD45+ CD3-CD19-NK1.1+ NKp46+ CD49a-CD49b+. For caspase activity assay, splenic NK cells were identified being CD45$^+$, lineage negative (CD3$^-$ CD19$^-$) NKp46$^+$ and stained as indicated by the manufacturer's protocol with Cell Event Caspase 3/7 Green Flow Cytometry Assay Kit (Molecular Probes). All samples were analyzed using BD Canto FACS instrument (BD Biosciences). Flow cytometry files were analyzed using FlowJo v10 software (Tree Star).

## Analysis of RNA

RNA levels from enriched NK cells were measured using quantitative reverse transcriptase PCR (qRT-PCR). RNA was isolated from cells using Qiagen RNeasy Plus Mini Kit (Qiagen). cDNA was synthesized using ~1 μg of DNase-treated RNA with random hexamers and SuperScript III reverse transcriptase (Invitrogen). qPCR of cDNA was performed using PowerSYBR Green PCR Master Mix (Applied Biosystems) using the following primers: *Bcl-2* (5'-CCTGTGGATGACTGAGTACCTG-3' and 5'-AGCC AGGAGAAATCAAACAGAGG-3'), *Bim* (5'-GGAGATACGGATTGCACAGGAG-3' and 5'-CTCCATAC CAGACGGAAGATAAAG-3'), *cytochrome c* (5'-GAGGCAAGCATAAGACTGGACC-3' and 5'-ACTC CATCAGGGTATCCTCTCC-3'), *HIF1a* (5'-CATCAGTTGCCACTTCCCCA-3' and 5'-GGCATCCAGAAG TTTTCTCACAC-3'), *hprt* (5'-CTGGTGAAAAGGACCTCTCGAAG-3' and 5'-CCAGTTTCACTAATGA CACAAACG-3'), *glut1* (5'-AAGAAGCTGACGGGTCGCCTCATGC-3' and 5'-TGAGAGGGACCA GAGCGTGGTG-3'), *pkm2* (5'-CAGGAGTGCTCACCAAGTGG-3' and 5'-CATCAAGGTACAGGCA CTACAC-3'), *hk2* (5'-GGAGAGCACGTGTGACGAC-3' and 5'-GATGCGACAGGCCACAGCA-3'), *gpi1* (5'-GTTGCCTGAAGAGGCCAGG-3' and 5'-GCTGTTGCTTGATGAAGCTGATC-3'), *ldha* (5'-CACAAG-CAGGTGGTGGACAG-3' and 5'-AACTGCAGCTCCTTCTGGATTC-3'), *pdk1* (5'-GATTCAGGTTCACGTC ACGCT-3' and 5'-GACGGATTCTGTCGACAGAG-3'), *Mcl-1* (5'-AGCTTCATCGAACCATTAGCAGAA-3' and 5'-CCTTCTAGGTCCTGTACGTGGA-3'), and *GAPDH* (5'-GTTGTCTCCTGCGACTTCA-3' and 5'-GGTGGTCCAGGGTTTCTTA-3'). qPCR was performed using StepOnePlus Real-Time PCR System (Applied Biosystems). Transcript abundance relative to GAPDH was calculated using delta-delta CT.

## Statistics

Graphed data are represented as means with standard error of the mean (SEM). Statistics were determined using GraphPad software, and one-way ANOVA was used when comparing multiple groups or unpaired Student's *t*-test when comparing two groups as indicated in the figure legends. Statistical significance indicated by n.s., no significant difference; *$p < 0.05$; **$p < 0.01$; ***$p < 0.001$; ****$p < 0.0001$.

## Additional information

### Funding

| Funder | Grant reference number | Author |
| --- | --- | --- |
| National Institute of Environmental Health Sciences | R35ES028365 | Gary Patti |
| National Institute of Allergy and Infectious Diseases | R01-AI131680 | Wayne M Yokoyama |

The funders had no role in study design, data collection and interpretation, or the decision to submit the work for publication.

### Author contributions

Francisco Victorino, Conceptualization, Data curation, Formal analysis, Investigation, Methodology, Project administration, Supervision, Visualization, Writing - original draft, Writing – review and editing; Tarin M Bigley, Data curation, Formal analysis, Investigation, Methodology, Writing – review and editing; Eugene Park, Data curation, Formal analysis, Investigation, Writing – review and editing; Cong-Hui Yao, Li-Ping Yang, Sytse J Piersma, Data curation, Formal analysis, Methodology, Writing – review and editing; Jeanne Benoit, Formal analysis, Methodology, Software, Writing – review and editing; Elvin J Lauron, Conceptualization, Investigation, Writing – review and editing; Rebecca M Davidson, Formal analysis, Funding acquisition, Methodology, Resources, Software, Supervision, Writing – review and

editing; Gary J Patti, Conceptualization, Funding acquisition, Investigation, Resources, Supervision, Writing – review and editing; Wayne M Yokoyama, Conceptualization, Funding acquisition, Project administration, Resources, Supervision, Writing – review and editing

### Author ORCIDs
Francisco Victorino (iD) http://orcid.org/0000-0001-7626-3219
Eugene Park (iD) http://orcid.org/0000-0002-2617-7571
Sytse J Piersma (iD) http://orcid.org/0000-0002-5379-3556
Gary J Patti (iD) http://orcid.org/0000-0002-3748-6193
Wayne M Yokoyama (iD) http://orcid.org/0000-0002-0566-7264

### Ethics
All of the animals were handled according to approved institutional animal care and use committee (IACUC) protocols (#20180293) of the University of Washington in St. Louis School of Medicine.

### Decision letter and Author response
Decision letter https://doi.org/10.7554/eLife.68484.sa1
Author response https://doi.org/10.7554/eLife.68484.sa2

## Additional files

### Supplementary files
• Transparent reporting form

### Data availability
Data generated or analyzed during this study has been deposited to the Dryad Digital Depository, available here: https://doi.org/10.5061/dryad.n5tb2rbvm.

The following dataset was generated:

| Author(s) | Year | Dataset title | Dataset URL | Database and Identifier |
| --- | --- | --- | --- | --- |
| Victorino F, Bigley TM, Park E, Yao CH, Benoit J, Yang LP, Piersma SJ, Lauron EJ, Davidson RM, Patti GJ, Yokoyama WM | 2021 | NK cell metabolic adaptation to infection promotes survival and viral clearance | https://doi.org/10.5061/dryad.n5tb2rbvm | Dryad Digital Repository, 10.5061/dryad.n5tb2rbvm |

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
