## [Decision Letter]

**Acceptance summary:**

This paper will be of interest for the readers in the field of viral immunology and metabolism. By using mice lacking hypoxia inducible factor-1α, the study unravels a previously unknown function of this factor in virus control by NK cells. It is shown that hypoxia inducible factor-1α supports survival of NK cells through efficient glucose metabolism required for optimal NK cell response to viral infection.

**Decision letter after peer review:**

Thank you for submitting your article "NK cell metabolic adaptation to infection promotes survival and viral clearance" for consideration by *eLife*. Your article has been reviewed by 3 peer reviewers, one of whom is a member of our Board of Reviewing Editors, and the evaluation has been overseen by Jos van der Meer as the Senior Editor. The following individual involved in review of your submission have agreed to reveal their identity: Laurent Brossay (Reviewer #2).

The reviewers have discussed their reviews with one another, and the Reviewing Editor has drafted this to help you prepare a revised submission. Although reviewers were enthusiastic with the presented results, they pointed to several issues that need to be addressed in the revised submission.

*Reviewer #1 (Recommendations for the authors):*

The presented findings are novel and relevant for the readers of *eLife*. The manuscript is well written and the conclusions are supported by the data. The experiments are overall well performed. However, given the modest differences in splenic ratio of IE1 and actin between 1αKO and wt mice, it would be better if the authors show MCMV titers in organs of HIF1α KO and control mice.

*Reviewer #2 (Recommendations for the authors):*

In this manuscript, the authors analyzed the role of HIF1a in NK cells in a variety of settings, including viral infection. HIF1a deficient NK cells appear to be mostly functional in terms of effector functions and ability to proliferate with only subtle differences with WT NK cells. This was also observed in HIF1a deficient Ly49H^+^ NK cells, yet in vivo Ly49H expansion is reduced in HIF1a KO mice. Response to IL-2 demonstrate that despite similar proliferation rate NK cell numbers were reduced indicating to the authors an NK cell survival issue. This was confirmed by measuring Bim and Bcl2, which were respectively decreased and increased. Increased cell death of HIF1a deficient NK cells during MCMV was confirmed. Mechanistically, the authors found that cell death was autophagy independent but due to an impaired glycolytic activity. The author concluded that in the absence of HIF1a, NK cells had an increase apoptosis due to abnormal glucose metabolism. Interestingly in tumor models two other reports (references #17 and 18) show that loss of HIF1a in NK cells is beneficial to the host. The authors try to speculate of why absence in HIF1a is beneficial in tumor models but detrimental during MCMV. Other differences were also noticed when compared to these 2 studies in vitro, which the authors attribute to culture conditions of NK cells. Overall, the experiments are well executed and are logical and the conclusions are supported by the data presented.

It is mentioned that the control mice used in this study do not express the transgene NcrCre and that instead floxed mice were used as controls. In general, it is recommended to use Cre mice as a control (see reference PMC5910175), but it is especially true for NKp46-iCre since expression of NKp46 an activating receptor is homozygote in the control while heterozygote in the ko. It would be reinsuring to show experiments with the appropriate controls especially given the differences with previous studies (references #17 and 18).

The differences with the 2 other studies are puzzling and the authors make reasonable assumptions and/or possible explanations. Because there is a direct NK cell activation during MCMV (M157/Ly49H) and presumably not in tumor models, could this explain the differences? Along the same line, how are the HIF1a deficient Ly49H negative NK cells behaving?

*Reviewer #3 (Recommendations for the authors):*

First paragraph of the introduction requires proofreading.

Part of the Methods referring to the Rag2-γRc mice transfer experiments is lacking.

The title does not highlight the contribution of HIF1a.

---

## [Author Response]

Reviewer #1 (Recommendations for the authors):The presented findings are novel and relevant for the readers of eLife. The manuscript is well written and the conclusions are supported by the data. The experiments are overall well performed. However, given the modest differences in splenic ratio of IE1 and actin between 1αKO and wt mice, it would be better if the authors show MCMV titers in organs of HIF1α KO and control mice.

We agree but our colony of HIF1a-floxed mice crossed to Ncr1-Cre mice is limited due to constraints of the pandemic. In order to address this comment, relevant conclusions require additional supporting data.

Reviewer #2 (Recommendations for the authors):It is mentioned that the control mice used in this study do not express the transgene NcrCre and that instead floxed mice were used as controls. In general, it is recommended to use Cre mice as a control (see reference PMC5910175), but it is especially true for NKp46-iCre since expression of NKp46 an activating receptor is homozygote in the control while heterozygote in the ko. It would be reinsuring to show experiments with the appropriate controls especially given the differences with previous studies (references #17 and 18).

We duly note and respect the reviewer’s comment about the observations made in PMC5910175 of off target effects using the Cre system. However, upon detailed examination of references #17 and #18, we found that our studies used identical controls as those papers (HIF1a-Flox mice). Therefore, the differences observed in references 17 and 18 compared to our studies are unlikely due to differences in these controls.

The differences with the 2 other studies are puzzling and the authors make reasonable assumptions and/or possible explanations. Because there is a direct NK cell activation during MCMV (M157/Ly49H) and presumably not in tumor models, could this explain the differences? Along the same line, how are the HIF1a deficient Ly49H negative NK cells behaving?

The reviewer asks whether or not ligation of Ly49H ligation during MCMV infection could explain differences observed in our studies compared to tumor models. Our data show that HIF1a expression is comparable between both Ly49H^+^ and Ly49H- NK cells during MCMV infection (Figure 1F), suggesting an “antigen”-independent effect but this does not rule out the possibility that HIF1a may be playing different roles in the two subsets. Ly49H^+^ NK cells dominate the MCMV response while Ly49H-negative subsets expand very little (Reference #6 Dokun et al.,). We provide new data here by evaluating Ly49H-negative NK cells, and found HIF1aKO-NK cell numbers were significantly reduced compared to Flox controls at day 3 and 7 but not at other time points (Figure 2E). In addition, we added more data looking at activation status and found a similar phenotype between Ly49H^+^ and Ly49H-negative NK cell subsets (Figure 3—figure supplement 1B, Figure 3—figure supplement 1C). We suspect that metabolic requirements of Ly49H-negative NK cells normalize similar to Ly49H-positive NK cells. Thus, our data indicates that HIF1a has a global impact on overall survival of NK cells independent of “antigen” specificity. Any further in depth analysis of NK cells to make relevant conclusions would require additional supporting data.

Reviewer #3 (Recommendations for the authors):First paragraph of the introduction requires proofreading.

As mentioned above, this text in paragraph 1 of the introduction was mistakenly shuffled at the end stages of manuscript preparation and has now been edited.

Part of the Methods referring to the Rag2-γRc mice transfer experiments is lacking.

This has now been added. “For homeostatic proliferation, 20x10^6^ CFSE labeled splenocytes from Flox and HIF1aKO mice were intravenously transferred into Rag-γRc mice. NK cell proliferation was measured from the spleen 4 days post transfer and immune cell analysis on day 14 post transfer.”

The title does not highlight the contribution of HIF1a.

Our new title is “HIF1α is required for NK cell metabolic adaptation during virus infection.”